# Optimizing the conservation of migratory species over their full annual cycle

Richard Schuster [1,2], Scott Wilson[1,3], Amanda D. Rodewald [4,5], Peter Arcese[6], Daniel Fink[4], Tom Auer [4] & Joseph.R. Bennett [1]

Limited knowledge of the distribution, abundance, and habitat associations of migratory species hinders effective conservation actions. We use Neotropical migratory birds as a model group to compare approaches to prioritize land conservation needed to support ≥30% of the global abundances of 117 species. Specifically, we compare scenarios from spatial optimization models to achieve conservation targets by: 1) area requirements for conserving >30% abundance of each species for each week of the year independently vs. combined; 2) including vs. ignoring spatial clustering of species abundance; and 3) incorporating vs. avoiding human-dominated landscapes. Solutions integrating information across the year require 56% less area than those integrating weekly abundances, with additional reductions when shared-use landscapes are included. Although incorporating spatial population structure requires more area, geographical representation among priority sites improves substantially. These findings illustrate that globally-sourced citizen science data can elucidate key trade-offs among opportunity costs and spatiotemporal representation of conservation efforts.

[1] Department of Biology, Carleton University, 1125 Colonel By Drive, Ottawa, ON K1S 5B6, Canada. [2] Ecosystem Science and Management Program, University of Northern British Columbia, 3333 University Way, Prince George, BC V2N 4Z9, Canada. [3] Environment and Climate Change Canada, National Wildlife Research Centre, 1125 Colonel By Drive, Ottawa, ON, Canada K1S 5B6. [4] Cornell Lab of Ornithology, 159 Sapsucker Woods Rd., Ithaca, NY 14850, USA. [5] Department of Natural Resources, Cornell University, Fernow Hall, #111, Ithaca, NY 14853, USA. [6] Department of Forest and Conservation Sciences, University of British Columbia, 2424 Main Mall, Vancouver, BC V6T 1Z4, Canada. Correspondence and requests for materials should be addressed to R.S. (email: richard.schuster@glel.carleton.ca)

Land-use change is a key threat to the conservation of bio-diversity, ecosystems[1], and the services they provide globally[2,3], and migratory species are particularly vulnerable to such change given the vast geographic areas they occupy over the annual cycle[4,5]. Indeed, a recent global assessment indicated that protected areas adequately protect the ranges of just 9% of migratory bird species[5]. Strategic approaches to identify and conserve habitats critical to the persistence of migratory species are therefore sorely needed.

Unfortunately, substantial gaps in knowledge of the abundance, distribution, and demography of most migratory species[6] have hampered strategic planning and led to uncertainty about the optimal allocation of conservation effort[5,7]. Given that populations of many migratory species continue to decline[4,8], there is an urgent need to identify portfolios of lands critical to the persistence of target species, and amenable to management in support of species conservation without compromising human well-being.

Multi-species decision support tools can facilitate the identification of areas crucial to the conservation of migratory species, but have remained intractable due to limits on knowledge and computing power. We capitalized on advances in models of bird species abundance and distribution using crowd-sourced data[9,10] and linear programing techniques[11] to develop a robust multi-species planning tool to estimate the land area needed to conserve 117 Nearctic-Neotropical migratory songbirds throughout the annual cycle (Supplementary Data 1). Specifically, we combined fine-scale, predictive models of distribution and abundance estimated weekly throughout the year with spatial optimization techniques[12] to identify the amount and type of land needed to reach our conservation targets given alternative planning scenarios at hemispheric scales.

We first estimated the abundance and distribution of 117 migratory bird species weekly, using spatiotemporal exploratory models[9,13] to calculate the relative abundance of each species throughout the annual cycle (Supplementary Movie 1). Incorporating information for each week of the year is especially important for migratory species, as this reflects their movements throughout the annual cycle and allows more precise estimates of their population distributions in space and time. We next recorded and compared the geographic area requirements and land cover types selected when optimizing for each week of the year independently and summing the total area over all weeks (hereafter, 'weekly'), versus optimizing over the entire year at once (hereafter, 'yearly'). Weekly optimizations for area efficiencies were developed to identify species-specific priorities for species at fine enough scales to capture short-term stopover sites. Our yearly approach optimized efficiency over the full annual cycle of each species, and emphasized temporal consistency in abundance hotspots more likely to reflect breeding and non-breeding regions. Because all existing conservation plans consider stationary phases of the breeding and non-breeding periods separately[14,15], our analysis is the first example of spatial optimization scenarios which track populations over their full annual cycle.

We next created area-optimized solutions designed to conserve lands used by 30% of the global populations of all 117 species in each of 52 weeks by sampling species (a) over their entire range, without accounting for spatial clustering of species abundance, or (b) by sampling within 5 regional population clusters, identified weekly to accommodate spatial clustering in species abundance. We wanted to account for the spatial clustering of species abundance because broadly distributed species often exhibit strong regional-scale variation in abundance across their range. Regional-scale variation in species' abundance may reflect a number of important processes affecting the ecology and

conservation of species, from variation in resource availability and land-use patterns to population-structure related to movement and migratory connectivity. By accounting for the spatial clustering of species abundance, the prioritization is stratified over multiple regions to ensure adequate protection over the entire species range. Because spatial clustering in species abundance—let alone its consequences for movement or connectivity—is poorly understood in most migratory species[16], we developed an innovative approach to account for structure statistically. Specifically, we used cluster analysis to delineate abundances into 5 spatial clusters of equal abundance and stratified our weekly sampling among clusters to capture the full geographic distribution of each species. Our use of five clusters is an example that minimized computational effort, but which could be extended to optimize by species or goal. Our 30% target is also arbitrary, but intermediate to the 17% of terrestrial ecosystems targeted by the Convention on Biodiversity[17] and 50% targets suggested by comparative analysis[18], and can be modified to reflect strategic goals[19].

Last, we compared area-based conservation plans designed to represent different perspectives about the potential contribution of human-modified lands to the conservation of migratory birds. Our 'intact habitat' approach emphasized the protection of relatively intact habitat as indicated by a low human footprint index[20] (Supplementary Fig. 1), whereas our 'shared-use' approach permitted the inclusion of landscapes converted to more intensive use by humans[21,22]. Our scenarios, termed intact habitat and permissive of shared-use, are analogous, but more general, than land sparing and land sharing scenarios. Exploring such constraints represents a critical step in conservation planning, given that human cultural history, values, and well-being can all affect conservation success and represent critical inputs into structured decisions about the most efficacious actions[23–25].

## Results

**Weekly vs. yearly approaches**. The land area required to achieve yearly conservation targets was 56% less on average than when area targets were summed across weeks (range = 49 to 65%; Table 1). Yearly solutions required relatively less land area than weekly approaches in shared-use scenarios (62% less) than in intact habitat scenarios (49% less, Table 1, Figs. 1, 2). Area reductions under yearly planning generally resulted from cases such as the inclusion of sites used by more than one species across two or more weeks of the annual cycle. A likely explanation for this difference is that the yearly approach will select sites that are used for longer periods of the annual cycle, and that there may be greater overlap of those areas between species than occurs in the short-term stopover sites included in the weekly approach, hence the larger area needed under the latter.

**Single populations vs. spatial clustering**. As expected, the area required to reach our conservation targets increased when we accommodated spatial clustering of species abundance, although relatively less so under a intact habitat (13% increase) compared to a shared-use (26% increase) scenario (Table 1, Figs. 1, 2). This reduction occurred in part because the homogenous cost structure used in our intact habitat scenario was less influential on site selection than was the heterogenous cost structure used in the shared-use scenario. Although we currently lack empirical data with which these spatial clusters can be validated, our predictions can be tested directly as tracking and genetic mapping techniques improve to allow comparisons of observed and predicted migration routes. That being noted, our current method provides a useful approach to

### Table 1 Area requirements to meet a 30% population target

| Area constraint | Single population weekly | Single population yearly | Spatial clustering of species abundance weekly | Spatial clustering of species abundance yearly |
|---|---|---|---|---|
| Shared-use | 14.38 | 5.51 | 20.03 | 6.93 |
| Intact habitat | 14.54 | 7.45 | 16.44 | 8.45 |

Prioritization approach results under shared-use and intact habitat scenarios are shown for 117 Neotropical migrant bird species. Table entries show the area needed to meet targets (million km^2). 'Weekly' prioritizes the most efficient target for each week of the year independently and sums the total area across all weeks. The yearly approach prioritized the most efficient target for all weeks combined. Single population identifies the 30% area target for each species from anywhere within the species range. Spatial clustering of species abundance identifies population sub-structure using a clustering approach to ensure representation from across the range of each species in each week of the year

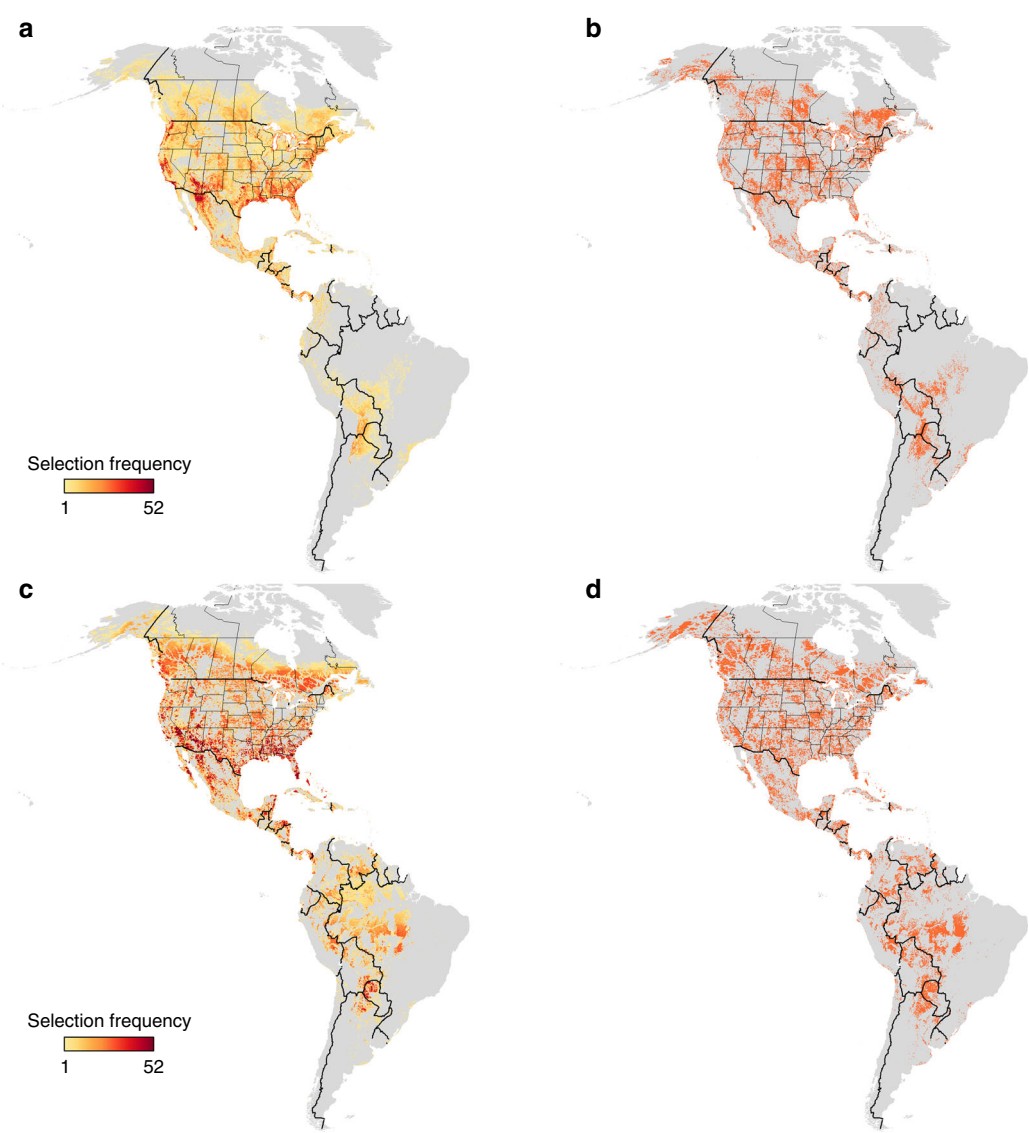

**Fig. 1** Single population comparison results. Areas prioritized for weekly and yearly planning under a shared-use approach allowing for the inclusion of human dominated landscapes vs. an intact habitat approach that excludes areas of high human footprint are shown. The prioritization is based on a target of 30% of global populations of 117 species of Neotropical migratory birds when each species range is considered as a single population. **a** = shared-use, weekly, **b** = shared-use, yearly, **c** = intact habitat, weekly, **d** = intact habitat, yearly. Panels **a** and **c** show how often (1, light yellow to 52, dark red) areas were selected across the weekly solutions. Panels **b** and **d** show whether an area was selected in a solution ( = red). Supplementary Fig. 2 presents a more detailed version of this figure focusing on northern South America

ensure geographic representation of spatial clustering of species abundance of a broad suite of species using publicly-available citizen science data in spatial planning tools.

Many conservation interventions, including land protection, are constrained by limits on fiscal or human resources and the opportunity costs of development. Our results show that sampling populations across the species range each week required almost twice the amount of land compared to yearly plans based on the relative abundance of species. Our work thus demonstrates the daunting problem of conserving sufficient land area such that migratory species' dynamic populations are protected throughout the year[5,7,10].

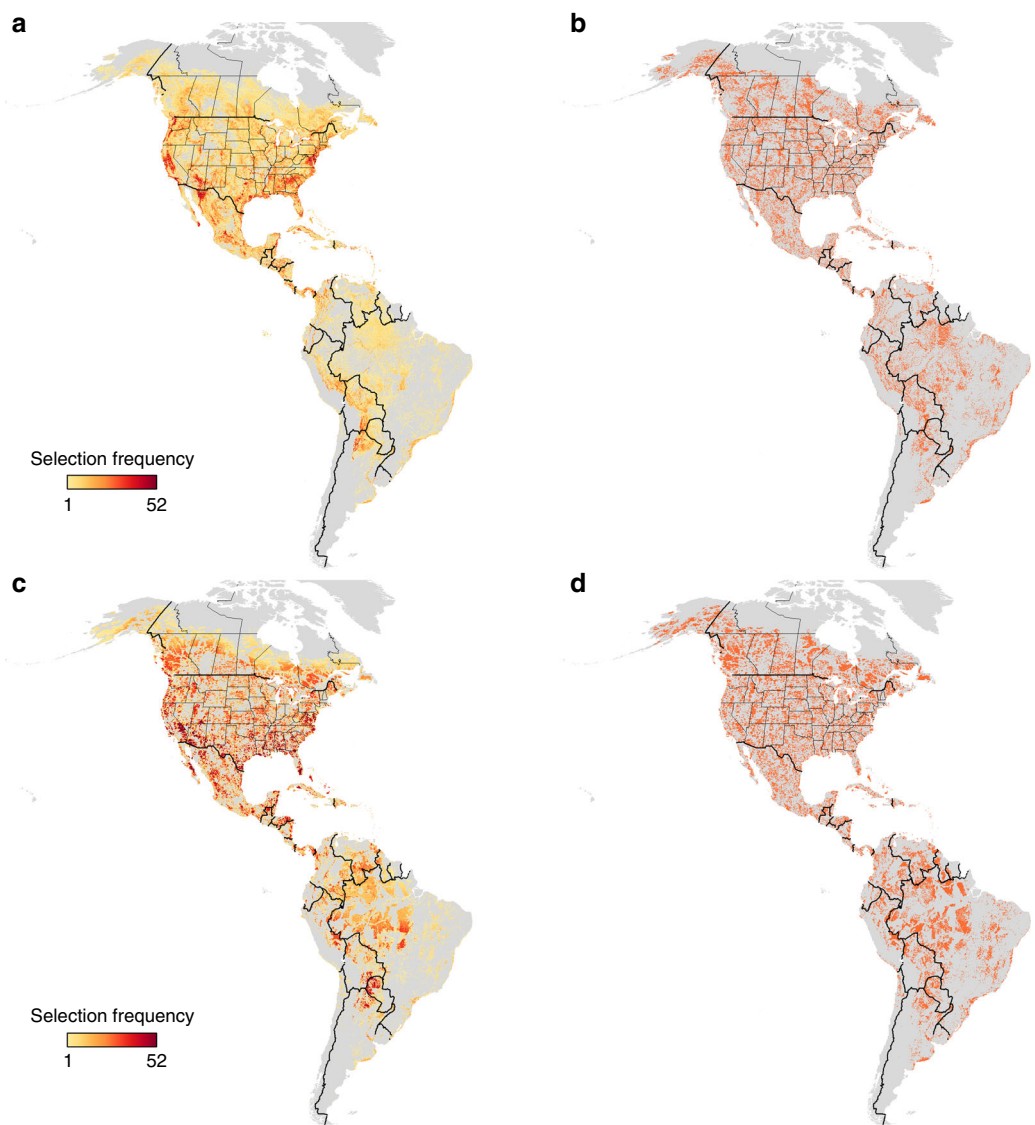

**Fig. 2** Spatial clustering comparison results. Spatial clustering of species abundance comparison of areas prioritized for weekly and yearly planning under a shared-use approach allowing for the inclusion of human dominated landscapes vs. an intact habitat approach that excludes areas of high human footprint. The prioritization is based on a target of 30% of global populations of 117 species of Neotropical migratory birds when each species range is considered with spatial clustering of species abundance (five regional clusters). **a** = shared-use, weekly, **b** = shared-use, yearly, **c** = intact habitat, weekly, **d** = intact habitat, yearly. Panels **a** and **c** show how often (1, light yellow to 52, dark red) areas were selected across the weekly solutions. Panels **b** and **d** show whether an area was selected in a solution ( = red). Supplementary Fig. 3 presents a more detailed version of this figure focusing on northern South America

**Intact habitat vs. shared-use**. Another key result of our work is that incorporating conservation objectives in human-dominated habitats may dramatically improve the efficiency of conservation area designs for migrants if their demographic performance is similar in 'working' and 'intact' landscapes. Under the yearly scenario we found that shared-use approaches required 26 and 18% less land area, respectively, than intact habitat approaches including or ignoring spatial clustering of species abundance (Table 1). We also found that intact habitat approaches selected different geographic areas and ecosystems than shared-use approaches. Most notably, intact habitat approaches selected larger areas of needle-leaved forest in boreal and mountainous zones of western Canada, and more broad-leaved evergreen forest in the eastern Andes and western Amazon basin (Figs. 1, 2; Table 2). Our findings thus add to a growing body of literature indicating the need to broaden the lens through which we view

conservation to both accommodate human livelihoods and conserve species[23–25].

**Overarching considerations**. Our findings suggest a need to re-evaluate conservation planning processes that are based on less precise methods. For example, government and non-governmental organizations allocate up to $1 billion annually to bird conservation based on aspatial targets and expert elicitation, with most actions directed to breeding habitat[14,15]. Our results suggest an alternative approach with a potential to meet conservation targets at lower land management cost and more compatible with human-dominated lands that can serve the dual purpose of supporting migratory species and human livelihoods. Overall, our results also illustrate potential trade-offs that conservation practitioners considering optimized portfolios must consider as additional targets and

**Table 2 Area selected (1000 km^2) for major land cover types**

| Land cover | Area available | Single population intact habitat | Single population shared-use | Spatial clustering of species abundance intact habitat | Spatial clustering of species abundance shared-use |
|---|---|---|---|---|---|
| Cropland/mosaic cropland | 2269 | 339 | 313 | 445 | 439 |
| Grassland | 5555 | 1198 | 1088 | 1313 | 1238 |
| Urban areas | 205 | 9 | 95 | 25 | 74 |
| Broadleaf deciduous forest | 1994 | 627 | 637 | 619 | 548 |
| Broadleaf evergreen forest | 6921 | 1595 | 735 | 2024 | 1433 |
| Needleleaf forest | 4599 | 1359 | 1006 | 1395 | 1160 |
| Mixed forest | 966 | 310 | 246 | 311 | 285 |
| Mosaic forest | 934 | 207 | 160 | 229 | 194 |
| Flooded forest | 540 | 148 | 97 | 162 | 136 |
| Shrubland | 4226 | 912 | 643 | 1135 | 864 |
| Wetland | 468 | 144 | 74 | 159 | 107 |
| Barren | 1053 | 207 | 79 | 208 | 109 |
| Total | 31,615 | 7055 | 5174 | 8025 | 6586 |

Values are based on yearly planning for shared-use vs. intact habitat scenarios and for single population vs. spatial clustering approaches. Area available is the total amount of each land cover available based on all cells throughout the year where > 1 species was present. Not all land cover classes are included in the table and therefore individual land cover values do not sum to the total in each column. Land cover data was extracted from the global land cover map for 2015 (300 m resolution)[55]. See Supplementary Table 1 for equivalent land area estimates under weekly planning scenario

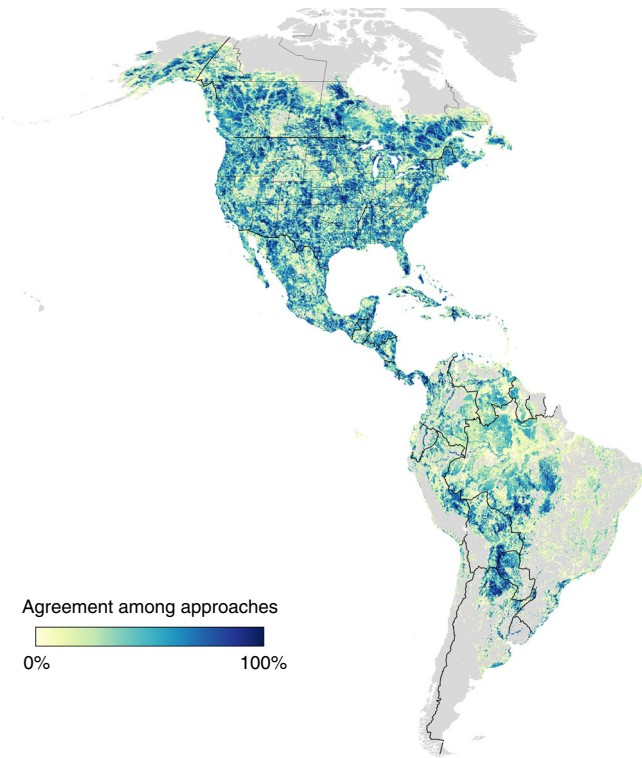

**Fig. 3** Range of agreement between the eight scenarios investigated. Darker blue indicates that most or all scenarios selected specific areas across the Western Hemisphere, and lighter yellow indicates areas of high scenario specificity. Scenario types considered: (i) summing scenarios for each species in each week of the year vs. optimizing over all weeks and species in a yearly approach, (ii) including vs. ignoring spatial clustering of species abundance, and (iii) incorporating vs. avoiding human-dominated landscapes in solutions

constraints are identified and incorporated in higher-level management models[25]. Even without consensus among conservation practitioners on focal scenarios, a considerable area of land was selected in at least six of the eight scenarios explored here,

illustrating that many priority areas meet a wide range of perspectives encapsulated in our scenarios (126, 000 km^2, Fig. 3). This indicates that even where approaches differ, conservation practitioners can use tools like the one described here to build consensus in hemispheric conservation efforts.

Several additional caveats arise from our results, particularly with respect to the shared-use and intact habitat scenarios. Implementing conservation action in working landscapes may be more challenging than in areas with less human activity if the opportunity costs of management are higher. For example, even if identified as a high-priority site for conservation in our shared-use scenarios, land already converted to human use may be more vulnerable to degradation in the future than more intact areas[26]. Such habitat degradation, especially if combined with other anthropogenic stressors that may directly or indirectly reduce survival or performance of wildlife[27], could make it difficult to reach population goals for species even if area needs are lower compared to less developed landscapes. In practice, both approaches are likely to be used given that target species will differ in their reliance on more or less developed habitat types[28]. Our prioritization scenarios provide planners with guidance on the approximate locations and requirements for land needed to meet our stated targets under a range of scenarios. With such portfolios in hand, planners can then more readily assess the cost-effectiveness of alternate approaches to land management and socio-economic policies most favorable to conservation and human well-being[23–25].

Additional species or landscape characteristics might also be considered in prioritization, depending on the mandate of conservation agencies. For example, the threat status of species has been incorporated into spatial planning recommendations[29,30]. Our analysis included 19 red or yellow-listed species by Partners-in-Flight (Supplementary Data 1). In addition, the specific life-history characteristics and habitat requirements of species, where available, can be important considerations. The 117 species in our analysis varied in spatiotemporal patterns of migration, habitat use, abundance, and population abundance. Although we did not address how variation in life history among species might influence results, it is clear that as high-resolution distribution models become available

for a larger range of migratory species, such as waterfowl, shorebirds, and marine, forest or grassland specialists, planners will have more opportunities to filter species or emphasize particular guilds to represent particular ecosystems or habitats. Different representation targets are also possible. We emphasize that the 30% target used here is illustrative only. In some cases, higher targets may be needed to avoid range contraction or the local extinction of sub-populations, to conserve ecological function such as seed dispersal or pest control[31], or to maintain the evolutionary potential of locally-adapted populations[32,33]. Nevertheless, our 30% target returned solutions in all cases which vastly exceed the areal extent of existing conservation plans in support of Neotropical migrant birds.

## Discussion

Ongoing declines in the abundance and distribution of many migratory species amid severe constraints on financial and human resources[34] point to an urgent need for area-based plans that achieve conservation targets while minimizing the opportunity costs of land conservation and impacts on human livelihood[23–25,35]. Our solutions minimized the total land area prioritized for conservation to provide an area-efficient portfolio of lands for further consideration by conservation planners. Four key lessons can be derived from our results. First, scenarios based on the distributions of abundance of all 117 species that were integrated over the entire year required less land area to meet conservation targets than scenarios based on optimizations that used the weekly distributions of those species throughout the year. This suggests difficult trade-offs between finer-scale spatiotemporal representation of habitats throughout the year, versus solutions that achieve the best compromise on population targets across the entire year. The latter appears more area-efficient, but may miss key stopover points along migratory routes. Second, accounting for spatial clustering of species abundance through stratified sampling across the entire distribution of species increased the total land area required to achieve conservation targets. Despite requiring more land area, ensuring geographic representation may be necessary to the long-term persistence of species, particularly in widely-distributed species with population genetic structure potentially reflecting adaptation to local conditions[32,33]. Third, area-based plans that accommodated human activity (shared-use) were generally more efficient than intact habitat approaches that avoided areas with a high human footprint. However, because migrants vary spatially and temporally in their tolerance of human-impacted landscapes[36], achieving conservation goals will likely require a portfolio of sites located in both intact and disturbed landscapes. Fourth, although our planning scenarios focused on Neotropical migratory birds, our approach could be easily adjusted and replicated in other migratory species and systems with sufficient data. In the case of birds, citizen science data and advanced prioritization tools allowed us to reveal marked efficiencies in area-based plans spanning the full annual cycle and multiple jurisdictions to conserve 117 individual species simultaneously.

## Methods

**Species selection**. We used the eBird citizen-science database (Sullivan et al., 2014) for this analysis. A total of 224 species were available and we identified a subset of these for analysis using the following procedure. We first examined annual eBird distribution maps for all species to identify Neotropical migratory species ($n = 181$ species), defined as those with breeding ranges in North America and non-breeding ranges that extend south of the Tropic of Cancer[37]. We then selected terrestrial passerines from this initial group ($n = 117$ species, see Supporting Information Data 1). These 117 species fell into two broad groups based on their breeding and stationary non-breeding ranges: (1) species where individuals breed in North America north of the US-Mexico border and migrate south of the Tropic of Cancer during the non-breeding period ($n = 101$ species, Supplementary

Data 1), and (2) species with both migratory and resident populations or sub-species, for which individuals from migratory populations north of the US-Mexico border move south of the Tropic of Cancer during the non-breeding period ($n = 16$ species). Both migratory and resident populations were included in our analysis.

**Approaches to conservation prioritization**. We created 8 planning scenarios using weekly STEM models for each of 117 focal species and incorporating different assumptions about temporal scale and cost metrics employed in prioritization. First, we created area-optimized solutions to conserve 30% of the global populations of all species by optimizing during each week of the year separately (i.e., conserving 30% of populations in each week and summing these individual solutions across the year) versus over the entire year (i.e., conserving 30% of total populations calculated once throughout the year, see below and Supplementary Fig. 4 for details). We next created area-optimized solutions to conserve 30% of the global populations of all species in each week by sampling each species a) over their entire range, without accounting for spatial clustering of species abundance, or b) as 5 regional population clusters identified weekly to accommodate spatial clustering of species abundance and migratory connectivity. Third, we compared area-based conservation plans designed to represent different perspectives about the potential contribution of human-modified landscapes to the conservation of migratory birds, while including either the unrestricted cost metric (each planning unit having the same cost of 1) or the human footprint cost metric (the planning unit cost equals the 2009, 1 km resolution human footprint metric by Venter et al.[20] to identify areas more and less subject to human use), to create a total of 8 scenarios (Supplementary Fig. 4). We used the prioritzr[38] R package for the analysis, which interfaces with the Gurobi[39] optimization software.

**Spatial prioritization approach**. Here we use the concept of systematic conservation planning[40], to inform choices about areas to protect, in order to optimize outcomes for biodiversity while minimizing societal costs[41]. To achieve the goal to optimize the trade-off between conservation benefit and socioeconomic cost, i.e., to get the most benefit for limited conservation funds, we strive to minimize an objective function over a set of decision variables, subject to a series of constraints. Integer linear programming (ILP) is the subset of optimization algorithms used here to solve reserve design problems. The general form of an ILP problem can be expressed in matrix notation as:

$$\text{Minimize } cx \text{ subject to } Ax \geq b \qquad (1)$$

where $x$ is a vector of decision variables (in our case, whether to prioritize an individual planning unit), $c$ and $b$ are vectors of known coefficients, and $A$ is the constraint matrix. In the minimum set cover problem, $c$ is a vector of costs for each planning unit, $b$ a vector of targets for each conservation feature, the relational operator would be ≥for all features, and $A$ is the representation matrix with $A_{ij} = r_{ij}$, the representation level of feature $i$ in planning unit $j$. We set an objective to find the solution that fulfills all the targets and constraints for the smallest area, which we use as our measure of cost[11]. This objective is similar to that used in Marxan, the most widely used spatial conservation planning tool[42], but has been shown to lead to more efficient solutions[11].

**Spatiotemporal exploratory models**. We used spatiotemporal exploratory models (STEM)[9,13,43] to generate estimates of relative abundance for each species. STEM is a type of species distribution model created as an ensemble of local regression models generated from a spatiotemporal block subsampling design. Repeatedly subsampling and partitioning the study extent into grids of spatiotemporal blocks, and then fitting independent regression models (base models) in each block produces an ensemble of partially overlapping local models. Estimates at a given location and date are made by averaging across all the local models that contain that location and date. Combining estimates across the ensemble controls for inter-model variability[44] and adapts to non-stationary predictor–response relationships[13].

The ensemble of spatiotemporal blocks was designed as a Monte Carlo sample of 100 randomly located spatiotemporal partitions of the study extent. This results in a uniformly distributed set of spatiotemporal blocks, and up to 100 local models covering each location in the study extent. To account for spatial variation in the density of the bird observation data[45], smaller spatiotemporal blocks (10° × 10° × 30 continuous days) were used north of 12° latitude and larger blocks (20° × 20° × 30 continuous days) were used in the southern portion of the study extent.

The bird observation data used to implement STEM came from the eBird citizen-science database[46]. The data included species counts from complete checklists collected under the 'traveling', 'stationary', and 'areal' protocols from January 1, 2004 to December 31, 2016 within the spatial extent bounded by 180° to 30° W Longitude (as well as Alaska between 150° E and 180° E). This resulted in a dataset consisting of 14 million checklists collected at 1.7 million unique locations, of which 10% were withheld for model validation.

Within each spatiotemporal block, species' occupancy and abundance were both assumed to be stationary. If there were at least 50 checklists with at least 10 detections of the given species within the spatiotemporal block, we fit a two-step boosted regression tree model designed to deal with zero-inflation[9] to predict the observed counts (abundance). If the minimum sample size requirements were not met for the spatiotemporal block, that spatiotemporal block was removed from the ensemble. The

boosted regression trees for both steps of the zero-inflation model were fit with the gbm package[47] with bag fraction = 0.80, learning rate or shrinkage = 0.05, and ntrees = 1000. The tree.depth parameter was set to 5 for the occurrence model and 10 for the abundance model, giving both models the ability to adapt to nonlinear and interacting predictor effects. We relied on the variance-reducing properties from averaging across the STEM ensemble to control for overfitting.

Three general classes of predictors were included in both boosted regression tree models: (i) spatial predictors to account for spatial (and spatiotemporal) patterns; (ii) temporal predictors to account for temporal variation at various scales; and (iii) predictors that describe the observation/detection process, which account for variation in detection rates, a nuisance when making inference about species occupancy and abundance.

Spatial information was captured using elevation[48] and NASA MODIS land[49] and water cover data. All 19 cover classes in the MODIS data were summarized within 2.8 × 2.8 km (784 ha) pixels centered at each eBird location. In each neighborhood, we computed the proportion of each class in the neighborhood (PLAND). To describe the spatial configuration of each class within each neighborhood we computed three statistics using FRAGSTATS[50] and SDMTools[51]: LPI an index of the largest contiguous patch, PD an index of the patch density, and ED an index of the edge density. Summarizing the land-cover information at the 2.8 × 2.8 km resolution reduced the impact of erroneous cover classifications, and reduced the impact of inaccurate eBird checklist locations. The time of day was used to model variation in availability for detection; e.g., diurnal variation in behavior, such as participation in the 'dawn chorus'[52]. Day of the year (1-366) was used to capture day-to-day changes in occupancy, and year was included to account for year-to-year differences. Finally, to account for variation in detection rates variables for the number of hours spent searching for species, the length of the transect traveled during the search, and the number of people in the search party were included in each base model.

Relative abundance was estimated as the expected count of birds of a given species on a standardized, hypothetical search conducted by a typical eBird participant starting from the center of the pixel from 7:00 to 8:00 a.m. while traveling 1 km. Estimates of relative abundance were rendered at weekly temporal resolution and 8.4 × 8.4 km spatial resolution and computed as the product of the estimated occupancy and, the estimated abundance conditional on occupancy, from the two steps of the boosted regression tree model.

Finally, to ensure good model performance, we required that the ensemble average computed for each relative abundance estimate have a sample size of at least 50 local models. In preliminary investigations based on expert review, we found that at least 50 local models were needed to control variation and limit extrapolation.

**Weekly and yearly approaches**. The two approaches for full annual cycle conservation prioritization used here, were (i) weekly, where one spatial prioritization problem was solved per week and the 52 results where summed together to create a final solution where each cell was selected between 0 (i.e., never) to 52 times (i.e., selected in each solution); (ii) yearly, where the features from each week were included in one spatial prioritization problem across all 52 weeks of the year. The solution of this scenario resulted in a cell either being selected or not. For the yearly approach, each week of the year is its own feature in the optimization. This means, that for a weekly scenario we would have 117 features (one for each species) and in the yearly 117*52 features (one for each species in each week of the year). To illustrate how the two approaches, differ, we have created a theoretical example for one species and four weeks (Supplementary Fig. 5). The example represents a very small study area (40 cells in total vs. 4.3 million in our actual analysis) and the species has a total of 100 individuals in each week. The target for the species is to protect 30% (i.e., 30 individuals) of the population in each week. The example shows that in general the weekly approach captures areas important for each week of the year, independent of other weeks of the year. The 'yearly' approach on the other hand optimizes across the year, thereby minimizing area requirements, but as a trade-off it does not necessarily capture the most important areas in each week of the year.

**Sampling for spatial clustering of species abundance**. Many of the species used here are represented by multiple sub-species or populations known or suspected to follow different migratory pathways and use different breeding or wintering habitats[5,19,53]. However, in the absence of detailed knowledge on migration pathways for the vast majority of species, we developed a system of stratified sampling to account for the weekly distribution and spatial structure of each of 117 focal species to ensure representation across their range throughout the annual cycle. To do so, we first conducted cluster analyses of predicted weekly distribution maps for all 117 species to identify 5 clusters of equal abundance that encompassed the entire species range to ensure representation across it. We sampled populations as 5 clusters because population structure information was missing for most species, to facilitate a data driven spatial delineation of population centers, and for computational efficiency. However, we note that the appropriate number of clusters among species may vary by species phylogeography or abundance should sufficient information become available. Our clustering approach re-draws the clusters in each week of the year, so does not consider whether spatial structure is maintained across weeks—it simply imposes a plausible structure within each week. Our cluster analysis was based on a dissimilarity matrix of geographic locations and abundances (which were weighted by 1/3 to primarily focus on

geographic effects and not bias cluster delineation toward spatially separated abundance clusters), and used the CLARA algorithm, which is an extension of the k-medoids technique for large datasets[54]. For an example see Supplementary Fig. 6.

**Land use constraints**. We used two metrics to constrain our systematic conservation prioritization. First, we used a constant, area-based cost metric for shared-use scenarios, whereby each planning unit was assigned a cost value of 1. In these scenarios, the optimization was driven solely by the species abundance predictions. Second, we used human footprint (2009; 1 km resolution)[20] to identify areas more and less subject to human use, access or development pressures; specifically, we calculated the mean human footprint value for each 8.4 × 8.4 km pixel in our study area and used it as the 'cost' of each pixel during prioritization. We used the human footprint metric as the cost metric in intact habitat scenarios. The human footprint layer represents a composite including the following human pressures: (1) the extent of built environments; (2) crop land; (3) pasture land; (4) human population density; (5) night-time lights; (6) railways; (7) roads; and (8) navigable waterways[20]. The human footprint values range from 0 to 50, with 0 representing no human pressures and 50 representing the highest human pressure possible. We used the human footprint metric as a continuous constraint in the prioritization approach.

**Land cover representation**. After the prioritization analyses, we summarized the major land cover types for each scenario that we generated. We used the 2015 data set of the global land cover map[55] at a 300 m resolution and clipped the original data to the study area. For each scenario, we used the geospatial data abstraction library[56] to warp the selected cells from the prioritization onto the raster grid of the land cover dataset. There were 37 land cover classes identified across scenarios and the frequency and area amount of each was summarized for all scenarios. As a final step we combined similar land cover classes into broader classes (Supplementary Table 2) and we used these to examine differences in area and land cover types selected under single season vs. full annual cycle planning and for intact habitat vs. shared-use scenarios (Table 2).

**Reporting summary**. Further information on experimental design is available in the Nature Research Reporting Summary linked to this article.

## Data availability

All data, computer code used in analysis, files generated from the analysis and outputs such as figures and tables have been deposited and are publicly available here: https://osf.io/58hgs/ (https://doi.org/10.17605/OSF.IO/58HGS).

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

## Acknowledgements

R.S. is supported by a Liber Ero Fellowship and Environment and Climate Change Canada (ECCC), A.D.R. by a Garvin endowment, and P.A. by the Natural Sciences and Engineering Research Council of Canada (NSERC), and J.R.B. by NSERC and ECCC. We also thank the eBird participants for their contributions and eBird team for their support. This work was funded by The Leon Levy Foundation, The Wolf Creek Charitable Foundation, NASA (NNH12ZDA001N-ECOF), Microsoft Azure Research Award (CRM: 0518680), and the National Science Foundation (ABI sustaining: DBI-1356308; computing support from CNS-1059284 and CCF-1522054).

## Author contributions

R.S., S.W., A.D.R., J.R.B. and P.A. conceived the study. R.S., D.F. and T.A. collected data and conducted analyses. All authors contributed to writing and editing the paper.

## Additional information

**Competing interests:** The authors declare no competing interests.

