## [Peer Review File · Nature Communications]

Reviewers' Comments:

Reviewer #1:

Remarks to the Author:

This paper provides a valuable advance to conservation planning for migratory species. Using bird abundance models, derived from eBird citizen science data, and linear programming the authors identify the portfolio of lands and habitats needed to conserve 30% of estimated population of 117 neotropical migrant passerine birds. This is the first paper I am aware of to use this approach. The paper is an important advance to recent work identifying how poorly existing global protected areas address the needs of migratory birds and calls for 'whole life-cycle' conservation planning to account for large geographic areas needed to support different life stages. The paper is generally well-written. The authors identify important trade-offs inherent in hemispheric conservation planning for migratory birds (optimizing conservation area for weekly versus annual abundance predictions, including population structure, and incorporating working lands – 'land-sparing' vs 'land-sharing') and analyze the effects on land area solutions (in land area needed to support 30% population goals: weekly optimization > annual optimization; population structure > no population structure; land-sparing > land-sharing) as well as those for combinations of trade-offs.

I think the paper could be improved by more strongly connecting the results to conservation practice as well as minor revisions for clarity.

A main conclusion of the paper is that conservation targets were achieved in 43% less land area in plans based on annual vs. weekly optimizations. This assumes that the goal is to minimize land area, and presumably cost, while still meeting the 30% abundance goal for all 117 species. Because conservation has been lacking an approach to address full life cycle needs of migratory birds, I think the more important conclusion may be that 2-3X more land area is required to meet the weekly habitat needs (and 30% goal) for these species. This would seem to account for critical stopover areas that would otherwise be missed by using the annual optimization.

The authors analyze 117 species of neotropical migratory birds categorized by migration pattern based on breeding and stationary non-breeding range. The broad range of species selected provides for a robust analysis with a diversity of migration patterns, habitat needs, abundance patterns and population structure. However, I was struck that little mention is made either in the analysis or in the discussion, conclusion and caveats, for species' conservation status – e.g. IUCN status, USFWS threatened or endangered status, declining species, species of special concern or other conservation-relevant categories. My rough count of the species list includes no fewer than 2 USFWS endangered species and 20+ threatened species or species of special concern. Listed status has the potential to drive conservation investment. I think species listed status category should be included in table 1 in supplemental materials and this is a trade-off worth mentioning in the conclusions, if not analyzing separately (e.g. how efficient is a land portfolio based on species of conservation concern in meeting the 30% target for all 117 species?).

The species list includes habitat generalists and specialists – could this be listed in the table 1 as well? To what extent does habitat specialization affect the optimization results?

The explanation of 'weekly' vs 'annual' in the paragraph beginning line 59 was not clear to me. The explanation in Table 1 legend was clear. I would recommend adapting that explanation for inclusion in the analysis paragraph that begins on line 59. I would have liked a figure that just compared 'weekly' vs annual 'separately', followed by one that includes population structure etc. Would be useful to have an example of the species for whose inclusion of sites used during one or two weeks of the year affected the differences between weekly and annual optimization. Are these perhaps stop-over or staging areas of conservation importance?

The consideration of population structure is novel, especially given our limited understanding of migratory connectivity. Population clusters were selected for each species based on weekly abundances – but why 5? Did this number of clusters emerge empirically from the distributions of each or most species? Does it relate to migration life cycle (e.g. breeding, migration stop-over, fall out areas and non-breeding areas)? The consideration of population structure in conservation planning is important – just needs more explanation for the delineation of clusters.

The land-sharing approach is more efficient but also has implicit assumptions about the long-term compatibility of lands within high human footprint in meeting the habitat needs for these species. Kremen and Merenlender's recent paper (Kremen, C., & Merenlender, A. M. (2018). Landscapes that work for biodiversity and people. *Science*, 362(6412), eaau6020.) would be good to cite in addition to Phalan et al. paper.

Reviewer #2:
Remarks to the Author:

This is an ambitious and novel study that seeks to map areas of conservation priority for a continental assemblage of migratory birds across the annual cycle. It uses sophisticated methods to draw detailed inferences about the spatiotemporal distributions of birds using citizen science data, and then examines a range of approaches to the 'minimum set' problem of conservation planning. I applaud the authors' efforts, and consider the idea behind the study to be an excellent and highly valuable one. However, the information provided in the Methods is insufficient to understand several key aspects of what was done, meaning that I actually found it quite hard to evaluate exactly what the authors were saying.

My primary concern is that the paper lacks a detailed description how the 'entire annual cycle' optimisation is actually done, and how this differs from the 'weekly' optimisation. This is central to the whole paper, but I'm afraid that despite having carefully read through the material provided, I still cannot fully grasp what the 'annual cycle' optimisation actually involves. Does it use an abundance metric that is averaged across all weekly maps for each species? Or does it sum them up across the year? I struggle to understand how a single 'all-year' distribution map could be used to indicate abundance for a species that is known to be migratory. This appears to be the authors' 'recommended' approach, given that it meets the conservation target in the smallest area, but I don't fully understand how it is calculated. Indeed, I actually found it quite hard to follow what the authors are really recommending in terms of 'best practice' methodology. If the full cycle approach is 'best' in terms of minimizing the area needed for conservation, does that genuinely mean it's the best at capturing the biological reality of what is needed by each species throughout the year? Or does the 'full cycle' approach exclude areas that might be critically important during short time windows (i.e. migration stopovers), which are captured by the weekly method? The paper desperately needs a detailed description of the differences between the weekly and full-cycle methods, together with a justification for why they method might be used.

The Methods also fail to provide sufficient information about how the 'sparing' and 'sharing' scenarios were implemented. Was a threshold used to exclude cells from the 'sparing' optimisation, or was 'human footprint' used as a linear constraint on the optimisation? In either case, I do not think that the 'sparing vs sharing' terminology is appropriate for the analyses presented in this paper. The sparing-sharing framework is explicitly concerned with measuring the trade-off between conservation and yield (of food or other commodities). Robust sparing-sharing comparisons therefore require that

the two strategies deliver exactly the same yield in a given scenario, but this is not the case here. This paper merely examines whether the area needed to conserve 30% of each species is larger/smaller if populations within human-dominated landscapes are brought under 'protection' alongside those within natural habitats (i.e. conservation seeks to ensure that current human land-uses within a given human-dominated area are maintained indefinitely). I suggest the authors use a different terminology to describe these comparisons, as this is not a robust sparing-sharing comparison in the absence of any explicit consideration of commodity yield.

Specific comments:

The Abstract is rather difficult to follow, and needs re-writing to be accessible to a wider academic audience and to get across the point of the study. It would help to briefly specify what 'different approaches to land prioritization' are being considered. L27 should specify the target is to conserve at least 30% of abundance – this isn't clear. Also not very clear what is meant by 'population structure' (spatial?) or 'representation' on L28-29. What does 'leading-edge abundance models' mean?

L62 – make clear here why it might be advantageous to use weekly optimization, rather than simply look at the full annual distribution – what is the rationale behind doing this?

L70-71 – I don't see how the cluster analysis allows you to 'accommodate spatial variation in population structure and migratory connectivity' – this needs to be re-worded to be more precise about what the clustering approach actually does. It does not account for migratory connectivity, as this term strictly refers to the maintenance of spatial population structure between seasons – the extent to which this occurs is unknown for the vast majority of species here. The clustering approach re-draws the clusters in each window, so does not consider whether spatial structure is maintained across windows – it simply imposes a plausible structure within each window.

L73 – if it can be 'easily modified', it would be great to see if the broad results change under a different threshold, eg 50% or higher...

L84-90 - The wording at the opening of the Results is a little confusing, partly because the term 'full annual cycle' is used to refer to the method based on the all-year distribution, whereas the 'weekly' method also captures the full annual cycle (and presumably does it in more detail). I think it would be better to use different terms for each of these scenarios. It might also help to re-phrase the key results in terms of the area needed to best protect all of the habitats utilised by populations throughout the annual cycle (i.e. deliver the conservation target), and then explain how/why the full cycle method allows this to be met in a smaller space.

L89-90 – I struggle to follow the meaning of this sentence, particularly the wording here: "... generally result from cases such as the inclusion of sites used by a single species in two or more weeks of the year, or by two or more species in during two or more weeks". Why not simply say 'used by one or more species in two or more weeks of the year'...?

L91 – what kind of 'population structure'? Be specific that this is spatial structure, as it could also refer to genetic or age structure.

L140 - Fig 3 does not get much discussion in the paper – as far as I can see it is only mentioned in this sentence, stating that there are some 'priority areas that most approaches agree on'. It would be useful to provide some more detail on where these areas are, and why. Moreover, I'm not convinced that the 'agreement' metric displayed in Fig 3 is particularly useful. Does it make sense to look at 'agreement' between sparing & sharing scenarios, given that some cells are effectively excluded from

one method (sparing) and not the other...? High agreement would therefore be limited to cells that were included in the 'sparing' scenarios (explaining why the agreement map more closely resembles the sparing maps than the sharing maps). Looking at agreement across scenarios effectively means you believe that all scenarios are equally useful / meaningful. Is that the case?

Figure 1 – are the color schemes comparable across all panels? What does the scale actually represent, and is it exactly the same for all panels? The text captions on b and d are abbreviated and not helpful.

Figure 2 – it is rather confusing that the legends for Figs 1 and 2 are almost identical. It would be better to re-word caption for Fig 2 to emphasize the key difference between this figure and Fig 1.

L236 – why are these species groupings even reported, given that the groupings are not used in the analysis? For those species with resident populations, how were these handled in the subsequent analysis? Were areas of residence included in the prioritization?

L247 – A detailed section is needed here to describe how the weekly vs annual cycle optimisations are done.

L254 – It would be helpful to provide brief descriptions of 'unrestricted cost metric' and 'human footprint cost metric' here.

L267 – what is 'x' in the case of this analysis? The abundance of each species in the cell?

L268 – what is the 'final term' here?

L268-269 – I don't follow what is meant here, please re-word: "... where relational operators for the constraint can be either \geq the coefficients."

L295 – how was model validation performed? Where are the results of this validation? Were any species excluded due to poor model performance?

L296 – what is meant by 'within each base model' here? Does it mean within each spatiotemporal block? Or the first time slice in each spatial block?

L299 – what kind of temporal trends? Within-year trends? Or trends across the years covered by the data?

L297 – some more detail is needed here on the BRT fitting within each spatiotemporal block (including the settings used eg bag fraction, learning rate, tree complexity etc). It would also be helpful to include a brief description of the types of relationships fitted by BRTs (linear / non-linear, interactions etc).

L303 – more detail needed on what is meant by 'spatial configuration' here – how many configuration variables, and what type?

L316 – why 50-100 base models? What determined the number used?

L317 – what determined whether abundance or occupancy was used? How were abundance and occupancy outputs combined in the prioritization analysis?

L316-318 – presumably the estimates of abundance for each species are each associated with a variance/uncertainty (propagated across the model ensemble), but these are essentially 'ignored' by taking the mean estimate forward in subsequent analysis. Was any model critique done to ensure that uncertainty levels were acceptable for each species & time slice?

L326 & 329 – ensure not insure

L328 – presumably cluster analysis was done using the model predicted distribution maps, now raw abundance data – perhaps best to make this clear here.

L334 – In what circumstances were adjustments to the default 5 clusters made, and how often? Why not pick a specific cluster number for each species based on the known number of subspecies/subpopulations?

L337 - Please explain the rationale for using the two different cost metrics here (i.e. why the constant cost metric was used, given that all cells were assigned the same value). I guess the 'constant cost' metric was used for land-sharing, and the other one used for land-sparing, but this isn't remotely clear. What was the threshold used for deciding whether a cell could be used in 'land-sparing' scenarios? Or was the human footprint used as a continuous constraint? The description of the differences between 'sparing' and 'sharing' need a much more detailed description.

Please find our responses to reviewer comments in bold below. Line numbers included refer to lines in text without tracked changes.

Reviewers' comments:

Reviewer #1 (Remarks to the Author):

This paper provides a valuable advance to conservation planning for migratory species. Using bird abundance models, derived from eBird citizen science data, and linear programming the authors identify the portfolio of lands and habitats needed to conserve 30% of estimated population of 117 neotropical migrant passerine birds. This is the first paper I am aware of to use this approach. The paper is an important advance to recent work identifying how poorly existing global protected areas address the needs of migratory birds and calls for 'whole life-cycle' conservation planning to account for large geographic areas needed to support different life stages. The paper is generally well-written. The authors identify important trade-offs inherent in hemispheric conservation planning for migratory birds (optimizing conservation area for weekly versus annual abundance predictions, including population structure, and incorporating working lands – 'land-sparing' vs 'land-sharing') and analyze the effects on land area solutions (in land area needed to support 30% population goals: weekly optimization > annual optimization; population structure > no population structure; land-sparing > land-sharing) as well as those for combinations of trade-offs.

Thank you for this encouraging comment.

I think the paper could be improved by more strongly connecting the results to conservation practice as well as minor revisions for clarity.

A main conclusion of the paper is that conservation targets were achieved in 43% less land area in plans based on annual vs. weekly optimizations. This assumes that the goal is to minimize land area, and presumably cost, while still meeting the 30% abundance goal for all 117 species. Because conservation has been lacking an approach to address full life cycle needs of migratory birds, I think the more important conclusion may be that 2-3X more land area is required to meet the weekly habitat needs (and 30% goal) for these species. This would seem to account for critical stopover areas that would otherwise be missed by using the annual optimization.

This is a good point and we have endeavoured to present a more balanced interpretation. To do so, we have thus substantially revised our discussion of the scenarios presented. Specifically, in line 115, we note that the greater land area required weekly scenario illustrates the daunting task of conserving dynamic populations across their entire distributions in space and time. In line 188 we also note that the more area-efficient scenario of annual integration may not capture all key areas along migratory pathways.

The authors analyze 117 species of neotropical migratory birds categorized by migration pattern based on breeding and stationary non-breeding range. The broad range of species selected provides for a robust analysis with a diversity of migration patterns, habitat needs, abundance patterns and population

structure. However, I was struck that little mention is made either in the analysis or in the discussion, conclusion and caveats, for species' conservation status – e.g. IUCN status, USFWS threatened or endangered status, declining species, species of special concern or other conservation-relevant categories. My rough count of the species list includes no fewer than 2 USFWS endangered species and 20+ threatened species or species of special concern. Listed status has the potential to drive conservation investment. I think species listed status category should be included in table 1 in supplemental materials and this is a trade-off worth mentioning in the conclusions, if not analyzing separately (e.g. how efficient is a land portfolio based on species of conservation concern in meeting the 30% target for all 117 species?).

Thank you for this comment. We have adjusted Table 1 in supplemental material accordingly and added Partners-in-Flight conservation assessment scores to the table. We have further modified the discussion section to discuss the trade-off mentioned by the reviewer (Line 158):

“Additional species or landscape characteristics might also be considered in prioritization, depending on the mandate of conservation agencies. For example, the threat status of species has been incorporated into spatial planning recommendations. Our analysis included 19 red or yellow-listed species by Partners-in-Flight (SI Table 1). In addition, the specific life-history characteristics and habitat requirements of species, where available, can be important considerations.”

The species list includes habitat generalists and specialists – could this be listed in the table 1 as well? To what extent does habitat specialization affect the optimization results?

This is an excellent point; however, given the many ways in which such issues might be explored, we believe that developing those issues here is beyond the scope of the current paper and risks diluting the more important messages and methods recognized by the reviewer's as substantial advances. We believe that once our Methods become widely-known, papers dedicated to these issues and their potential consequences for management and planning outcomes will get published. Specifically, we would suggest that 'generalist – specialist' distinctions are necessarily subjective, given any number of life history traits might offer potentially interesting or practical influences on results, depending on goals. Thus, we elected not to add this additional information to SI Table 1. However, in an effort to address the reviewer's important point positively, we have added text to Discussion at Line 163, to make it clearer that, as more data become available, planners will have an increasing ability to apply a range of filters to tailor their prioritisations to particular goals by targeting focal taxa. The added text and sentence above are intended to make these points succinctly (added text follows): “The 117 species in our analysis varied in spatiotemporal patterns of migration, habitat use, abundance, and population abundance. Although we did not address how variation in life history among species might influence results, it is clear that as high-resolution distribution models become available for a larger range of migratory species, such as waterfowl, shorebirds, and marine, forest or grassland specialists, planners will have more opportunities to filter species or emphasize particular guilds to represent particular ecosystems or habitats.”

The explanation of 'weekly' vs 'annual' in the paragraph beginning line 59 was not clear to me. The explanation in Table 1 legend was clear. I would recommend adapting that explanation for inclusion in the analysis paragraph that begins on line 59.

Thank you for this comment. We have adjusted text to better reflect what was presented in Table 1 legend (Line 65; text follows):

“We next recorded and compared the geographic area requirements and land cover types selected when optimizing for each week of the year independently and summing the total area across all weeks (hereafter, “weekly”), versus combining the data for all weeks over the entire annual cycle (hereafter, “integrated annual”).”

In addition, we have also added a supplemental figure to better explain these scenarios.

I would have liked a figure that just compared ‘weekly’ vs annual ‘separately’, followed by one that includes population structure etc.

If we understand the reviewer comment correctly, Figure 1 and 2 do show this, by presenting the four scenarios (1. Weekly, sharing; 2. Annual, sharing; 3. Weekly, sparing; 4. Annual, sparing) without using population structure (Fig 1) and then in Figure 2 the scenarios with population structure. We have tried to clarify this in figure legends. However, we remain willing to adjust text to address the reviewer’s comments differently if we’ve mis-understood the request.

Would be useful to have an example of the species for whose inclusion of sites used during one or two weeks of the year affected the differences between weekly and annual optimization. Are these perhaps stop-over or staging areas of conservation importance?

We thank the reviewer for this comment, which we understand given the complexity of the technical issues being addressed and applied. To address this question as definitively as possible, we have taken the reviewer’s suggestion and created a new, supplementary figure and detailed legend which we hope addresses these questions clearly and explicitly. To do so, we have developed a theoretical example to illustrate the step by step processed undertaken in our ‘weekly’ versus ‘integrated annual’ approach as SI Fig 6. We have included our comprehensive figure legend below, which explains the examples provided and their consequences for representation of population structure in each case, and which was applicable to our very larger prioritization problem we solve in this paper.

The new legend and supplemental figure follow:

“Schematic illustration depicting the differences between the weekly and ‘integrated annual’ approach we used. The example distribution data in the four top panels of a) and b) shows the species abundance distribution for a theoretical species over four weeks. The darker the green the more individuals are present. In the weekly approach a) each week is run through the spatial prioritization approach individually, resulting in a solution for each week. The bottom figure in a) shows those four maps summed together, highlighting the areas of highest abundance (dark green) for each individual week. In the integrated annual approach b) the distributions from the four weeks enter one spatial prioritization approach at the same time, but as distinct features, meaning they are not combined (e.g. summed) in any way before the prioritization is run. The resulting figure (bottom of b)) shows a gain in area efficiency (i.e. requires less area to meet the 30% target used in both approaches), but this comes at the cost of not capturing areas of highest abundance (dark green) for three out of four weeks (week 1-3). The reason why the ‘integrated annual’ approach is not capturing the areas of

highest abundance for weeks 1-3 is that the algorithm is trying to maximize the co-benefits for all features (i.e. weeks), while minimizing cost, an approach termed complementarity. As a result, the solution does not focus on areas of high abundance alone (as the single feature approaches in a) did), but instead identifies areas that can be selected, which would be beneficial for multiple features/weeks at the same time. As an example, the 30% target for week 1 is covered by the 3 top left cells in the bottom figure of b, but this does not capture the areas of highest abundance for a (moving) species, but might get the leading or trailing edge in some cases because of the integration with other weeks, as shown here. This relatively simple example, using only 40 planning units per week, shows that realized co-benefits between features can look very different from approaches that try to maximize the benefit for a single feature/week.”

The consideration of population structure is novel, especially given our limited understanding of migratory connectivity. Population clusters were selected for each species based on weekly abundances – but why 5? Did this number of clusters emerge empirically from the distributions of each or most species? Does it relate to migration life cycle (e.g. breeding, migration stop-over, fall out areas and non-breeding areas)? The consideration of population structure in conservation planning is important – just needs more explanation for the delineation of clusters.

We appreciate the reviewer’s recognition of the novelty of our approach, which was made possible by the fine-scale spatiotemporal mapping of species abundance adopted here. We have now made clearer, however, that the choice of 5 was arbitrary, and for computational efficiency. We have added text to Methods (Line 400) to clarify this decision, its consequences and extensions:

“We sampled populations as 5 clusters because population structure information was missing for most species, to facilitate a data driven spatial delineation of population centers, and for computational efficiency. However, we note that the appropriate number of clusters among species may vary by species phylogeography or abundance should sufficient information become available. Our clustering approach re-draws the clusters in each week of the year, so does not consider whether spatial structure is maintained across weeks – it simply imposes a plausible structure within each week.”

The land-sharing approach is more efficient but also has implicit assumptions about the long-term compatibility of lands within high human footprint in meeting the habitat needs for these species. Kremen and Merenlender’s recent paper (Kremen, C., & Merenlender, A. M. (2018). Landscapes that work for biodiversity and people. *Science*, 362(6412), eaau6020.) would be good to cite in addition to Phalan et al. paper.

We thank the reviewer for this suggestion and have added a reference to Kremen & Merenlender.

Reviewer #2 (Remarks to the Author):

This is an ambitious and novel study that seeks to map areas of conservation priority for a continental assemblage of migratory birds across the annual cycle. It uses sophisticated methods to draw detailed inferences about the spatiotemporal distributions of birds using citizen science data, and then examines a range of approaches to the ‘minimum set’ problem of conservation planning. I applaud the authors’ efforts, and consider the idea behind the study to be an excellent and highly valuable one.

Thank you for this encouraging comment.

However, the information provided in the Methods is insufficient to understand several key aspects of what was done, meaning that I actually found it quite hard to evaluate exactly what the authors were saying.

We have tried very hard to address these concerns below.

My primary concern is that the paper lacks a detailed description how the ‘entire annual cycle’ optimisation is actually done, and how this differs from the ‘weekly’ optimisation. This is central to the whole paper, but I’m afraid that despite having carefully read through the material provided, I still cannot fully grasp what the ‘annual cycle’ optimisation actually involves. Does it use an abundance metric that is averaged across all weekly maps for each species? Or does it sum them up across the year? I struggle to understand how a single ‘all-year’ distribution map could be used to indicate abundance for a species that is known to be migratory.

Thank you very much for this comment. We have added SI Fig. 6 as introduced above, as well as adding the following section to Methods to make the differences clearer (Line 376):

Weekly and integrated annual approaches

The two approaches for full annual cycle conservation prioritization used here, were i) weekly, where one spatial prioritization problem was solved per week and the 52 results were summed together to create a final solution where each cell was selected between 0 (i.e. never) to 52 times (i.e. selected in each solution); ii) integrated annual, where the features from each week were included in one spatial prioritization problem across all 52 weeks of the year. The solution of this scenario resulted in a cell either being selected or not. To illustrate how the two approaches, differ, we have created a theoretical example for one species and four weeks (SI Fig. 6). The example represents a very small study area (40 cells in total vs 4.3 million in our actual analysis) and the species has a total of 100 individuals in each week. The target for the species is to protect 30% (i.e. 30 individuals) of the population in each week. The example shows that in general the weekly approach captures areas important for each week of the year, independent of other weeks of the year. The ‘integrated annual’ approach on the other hand optimizes across the year, thereby minimizing area requirements, but as a trade-off it does not necessarily capture the most important areas in each week of the year.

This appears to be the authors’ ‘recommended’ approach, given that it meets the conservation target in the smallest area, but I don’t fully understand how it is calculated. Indeed, I actually found it quite hard to follow what the authors are really recommending in terms of ‘best practice’ methodology. If the full cycle approach is ‘best’ in terms of minimizing the area needed for conservation, does that genuinely mean it’s the best at capturing the biological reality of what is needed by each species throughout the year? Or does the ‘full cycle’ approach exclude areas that might be critically important during short time windows (i.e. migration stopovers), which are captured by the weekly method? The paper desperately needs a detailed description of the differences between the weekly and full-cycle methods, together with a justification for why they method might be used.

This is a good point, and we agree that a more balanced interpretation was required. We have thus substantially revised our discussion of these scenarios. Specifically, in line 115, we note that the greater land area required weekly scenario illustrates the daunting task of conserving dynamic populations across their entire distributions in space and time. In line 188 we also note that the more area-efficient scenario of annual integration may not capture all key areas along migratory pathways.

We have added SI Fig. 6 as introduced above, as well as adding a section to Methods to make the differences clearer (Line 376): see previous comment for text

The Methods also fail to provide sufficient information about how the 'sparing' and 'sharing' scenarios were implemented. Was a threshold used to exclude cells from the 'sparing' optimisation, or was 'human footprint' used as a linear constraint on the optimisation?

Thank you for pointing this out.

We have provided more details on the human footprint metric and its use in the 'Land use constraints' section of Methods (Line 419):

"We used the human footprint metric as the cost metric in low-impact scenarios. The human footprint layer represents a composite including the following human pressures: (1) the extent of built environments; (2) crop land; (3) pasture land; (4) human population density; (5) night-time lights; (6) railways; (7) roads; and (8) navigable waterways. The human footprint values range from 0 to 50, with 0 representing no human pressures and 50 representing the highest human pressure possible. We used the human footprint metric as a continuous constraint in the prioritization approach."

We have also clarified our terminology around these scenarios. Please see response to comment below.

In either case, I do not think that the 'sparing vs sharing' terminology is appropriate for the analyses presented in this paper. The sparing-sharing framework is explicitly concerned with measuring the trade-off between conservation and yield (of food or other commodities). Robust sparing-sharing comparisons therefore require that the two strategies deliver exactly the same yield in a given scenario, but this is not the case here. This paper merely examines whether the area needed to conserve 30% of each species is larger/smaller if populations within human-dominated landscapes are brought under 'protection' alongside those within natural habitats (i.e. conservation seeks to ensure that current human land-uses within a given human-dominated area are maintained indefinitely). I suggest the authors use a different terminology to describe these comparisons, as this is not a robust sparing-sharing comparison in the absence of any explicit consideration of commodity yield.

We thank the reviewer for their comment. We have changed the terminology as the reviewer suggested. We replaced sharing with 'shared-use' and sparing with 'low-impact' throughout the text.

We have further included an additional sentence at the end of intro to put our terms into context of sharing-sparing (Line 88):

"Our scenarios, termed low-impact and permissive of shared-use, are analogous, but more general, than sparing and sharing scenarios."

Specific comments:

The Abstract is rather difficult to follow, and needs re-writing to be accessible to a wider academic

audience and to get across the point of the study. It would help to briefly specify what ‘different approaches to land prioritization’ are being considered. L27 should specify the target is to conserve at least 30% of abundance – this isn’t clear. Also not very clear what is meant by ‘population structure’ (spatial?) or ‘representation’ on L28-29. What does ‘leading-edge abundance models’ mean?

We appreciate the reviewer’s concerns and have considerably revised the abstract to improve clarity and flow, but we of course remain open to editorial suggestions.

L62 – make clear here why it might be advantageous to use weekly optimization, rather than simply look at the full annual distribution – what is the rationale behind doing this?

We have added a sentence to help clarify why weekly information is beneficial over annual distributions (Line 62):

“Incorporating information for each week of the year is especially important for migratory species, as this reflects their movements throughout the annual cycle and allows more precise estimates of their population distributions in space and time.”

L70-71 – I don’t see how the cluster analysis allows you to ‘accommodate spatial variation in population structure and migratory connectivity’ – this needs to be re-worded to be more precise about what the clustering approach actually does. It does not account for migratory connectivity, as this term strictly refers to the maintenance of spatial population structure between seasons – the extent to which this occurs is unknown for the vast majority of species here. The clustering approach re-draws the clusters in each window, so does not consider whether spatial structure is maintained across windows – it simply imposes a plausible structure within each window.

Thank you very much for this observation. We have removed the term migratory connectivity from the section in question, as well as throughout the text, where ‘variation in population structure and migratory connectivity’ was used. To address the reviewer’s recommendation to better reflect what we actually did with the clustering approach, we are now use the term “spatial population abundance variation”. We have further added the following sentence to the Methods section (Line 404) to clarify what our method produced:

“Our clustering approach re-draws the clusters in each week of the year, so does not consider whether spatial structure is maintained across weeks – it simply imposes a plausible structure within each week.”

L73 – if it can be ‘easily modified’, it would be great to see if the broad results change under a different threshold, eg 50% or higher...

Thank you for this comment. We have adjusted the sentence in question to illustrate that we are focusing on one set of targets, but if different goals are stated, the approach could be adjusted to examine the consequences of choosing different goals (Line 81):

“Our 30% target is also arbitrary, but intermediate to the 17% of terrestrial ecosystems targeted by the Convention on Biodiversity and 50% targets suggested by comparative analysis, and can be modified to reflect strategic goals.”

L84-90 - The wording at the opening of the Results is a little confusing, partly because the term 'full annual cycle' is used to refer to the method based on the all-year distribution, whereas the 'weekly' method also captures the full annual cycle (and presumably does it in more detail). I think it would be better to use different terms for each of these scenarios. It might also help to re-phrase the key results in terms of the area needed to best protect all of the habitats utilised by populations throughout the annual cycle (i.e. deliver the conservation target), and then explain how/why the full cycle method allows this to be met in a smaller space.

Thank you for this comment. We have changed our terminology from “full annual cycle” to “integrated annual” to better represent the distinction between this and the weekly scenarios. We further have added an illustration (SI Fig. 6) to the paper to help with the conceptual difference between weekly and integrated annual approaches, as both reviewer’s have raised concerns about clarifying the distinction between the two.

We have further substantially revised our discussion of these scenarios. Specifically, in line 115, we note that the greater land area required weekly scenario illustrates the daunting task of conserving dynamic populations across their entire distributions in space and time. In line 188 we also note that the more area-efficient scenario of annual integration may not capture all key areas along migratory pathways.

L89-90 – I struggle to follow the meaning of this sentence, particularly the wording here: “... generally result from cases such as the inclusion of sites used by a single species in two or more weeks of the year, or by two or more species in during two or more weeks”. Why not simply say ‘used by one or more species in two or more weeks of the year’...?

We changed the wording to the recommendation of the reviewer.

L91 – what kind of ‘population structure’? Be specific that this is spatial structure, as it could also refer to genetic or age structure.

We thank the reviewer for pointing out the needed clarification. We have now revised our terminology in text to ‘population abundance’ from ‘population structure’ to: 1) more precisely emphasize that our sampling procedure was based on spatial variation in the estimated abundance of species; and 2) to avoid confusion with the more general term, encompassing population genetic structure, age structure, or other factors not addressed here.

L140 - Fig 3 does not get much discussion in the paper – as far as I can see it is only mentioned in this sentence, stating that there are some ‘priority areas that most approaches agree on’. It would be useful to provide some more detail on where these areas are, and why. Moreover, I’m not convinced that the ‘agreement’ metric displayed in Fig 3 is particularly useful. Does it make sense to look at ‘agreement’ between sparing & sharing scenarios, given that some cells are effectively excluded from one method (sparing) and not the other...? High agreement would therefore be limited to cells that were included in the ‘sparing’ scenarios (explaining why the agreement map more closely resembles the sparing maps

than the sharing maps). Looking at agreement across scenarios effectively means you believe that all scenarios are equally useful / meaningful. Is that the case?

Thank you very much for this comment. In light of the reviewer comment, we have adjusted Figure 3 to better illustrate our point, and to modify the section in question (Line 136) to better reflect this clarification:

“Even without consensus among conservation practitioners on which scenario to focus on, there is still a considerable amount of land selected in at least six of the eight scenarios investigated, illustrating potential focal areas that could be compatible with the various perspectives encapsulated by our scenarios (126, 000 km², Figure 3). This serves to illustrate that even if there are differences among approaches conservation practitioners might favor, we can build cross approach consensus results that can serve as starting points in hemispheric conservation efforts.”

Figure 1 – are the color schemes comparable across all panels? What does the scale actually represent, and is it exactly the same for all panels? The text captions on b and d are abbreviated and not helpful.

We have removed the text captions from panels b and d now, but added text to figure legend to help clarify color schemes and what the values represent:

“Panels a) and c) show how often (1, light yellow to 52, dark red) areas were selected across the weekly solutions. Panels b) and d) show whether an area was selected in a solution (= red).”

Figure 2 – it is rather confusing that the legends for Figs 1 and 2 are almost identical. It would be better to re-word caption for Fig 2 to emphasize the key difference between this figure and Fig 1.

We have now started the legends for both figures with the differences between the two. Figure 1 now starts:

“Single population comparison of areas prioritized for weekly ...”

and Figure starts with:

“Spatial population abundance variation comparison of areas prioritized for weekly ...”

L236 – why are these species groupings even reported, given that the groupings are not used in the analysis? For those species with resident populations, how were these handled in the subsequent analysis? Were areas of residence included in the prioritization?

We felt it was important to include information on those two groupings for interested readers. We have added a sentence to the species selection section in methods to clarify how we dealt with resident populations (Line 269): “Both migratory and resident populations were included in our analysis.”

L247 – A detailed section is needed here to describe how the weekly vs annual cycle optimisations are done.

Thank you very much for this comment. We have added SI Fig. 6 as introduced above, as well as adding the following section to Methods to make the differences clearer (Line 376):

“Weekly and integrated annual approaches

The two approaches for full annual cycle conservation prioritization used here, were i) weekly, where one spatial prioritization problem was solved per week and the 52 results were summed together to create a final solution where each cell was selected between 0 (i.e. never) to 52 times (i.e. selected in each solution); ii) integrated annual, where the features from each week were included in one spatial prioritization problem across all 52 weeks of the year. The solution of this scenario resulted in a cell either being selected or not. To illustrate how the two approaches differ, we have created a theoretical example for one species and four weeks (SI Fig. 6). The example represents a very small study area (40 cells in total vs 4.3 million in our actual analysis) and the species has a total of 100 individuals in each week. The target for the species is to protect 30% (i.e. 30 individuals) of the population in each week. The example shows that in general the weekly approach captures areas important for each week of the year, independent of other weeks of the year. The ‘integrated annual’ approach on the other hand optimizes across the year, thereby minimizing area requirements, but as a trade-off it does not necessarily capture the most important areas in each week of the year.”

L254 – It would be helpful to provide brief descriptions of ‘unrestricted cost metric’ and ‘human footprint cost metric’ here.

We have modified the section in question to include a brief description of the cost metrics (Line 283):
“Third, we compared area-based conservation plans designed to represent different perspectives about the potential contribution of human-modified landscapes to the conservation of migratory birds, while including either the unrestricted cost metric (each planning unit having the same cost of 1) or the human footprint cost metric (the planning unit cost equals the 2009, 1 km resolution human footprint metric by Venter et al. to identify areas more and less subject to human use), to create a total of 8 scenarios (SI Fig. 3).”

L267 – what is ‘x’ in the case of this analysis? The abundance of each species in the cell?

We have clarified the definition of x (Line 301):

“Where x is a vector of decision variables (in our case, whether to prioritize an individual planning unit)”

L268 – what is the ‘final term’ here?

We removed the following sentence as it was confusing and did not discuss potential variations to the \geq sign (which are not relevant for this analysis):

“The final term specifies a series of structural constraints where relational operators for the constraint can be either \geq the coefficients.”

L268-269 – I don’t follow what is meant here, please re-word: “... where relational operators for the constraint can be either \geq the coefficients.”

Please see comment above: This comment refers to the basic (most flexible) formulation of the ILP problem, wherein it is not necessary to include the ‘≥’ operator, given that ‘=’ or ‘≤’ are also possible. However, because this aspect is a distraction in our case, we removed the sentence and use ≥ in the formula, as suggested.

L295 – how was model validation performed? Where are the results of this validation? Were any species excluded due to poor model performance?

To assess model quality, we validated the model’s ability to predict the observed patterns of occupancy and abundance using independent validation data. The statistics were evaluated using a Monte Carlo design of 25 spatially balanced samples to help control for the uneven spatial distribution of the validation data (Fink et al. 2010; Roberts et al. 2017). To quantify the predictive performance we used the Area Under the Curve (AUC) and Kappa (Cohen 1960) statistics to describe the models’ ability to classify un/occupied sites (Freeman & Moisen 2008). AUC measures a model’s ability to discriminate between positive and negative observations (Fielding & Bell 1997) as the probability that the model will rank a randomly chosen positive observation higher than a randomly chosen negative one. Cohen’s Kappa statistic (Cohen 1960) was designed to measure classification performance accounting for the background prevalence. To quantify the quality of the occupancy estimate as a probability, we evaluated AUC and Kappa. To quantify the quality of the abundance estimates we computed Spearman’s Rank Correlation (SRC) and the percent Poisson Deviance Explained (P-DE). SRC measures how well the abundance estimates rank the observed abundances and the P-DE measures the correspondence between the magnitude of the estimated counts and observed counts. All of the metrics discussed above were evaluated separately each species, for each week of the year, across the extent of its modeled range at a spatial scale of 8.4km.

The approach we took to ensure good model performance was based on an informal preliminary study using expert review to set minimum regional sample size requirements. The goal was to find a minimum sample size that would guarantee mostly accurate predictions within regions with sufficient data density and strong extrapolations were mostly excluded in regions of low data density. Operationally, we required that at least 50 of the 100 spatiotemporal blocks covering any given location and time have 1) 50 or more checklists and 2) 10 or more detections of the given species with the block. Practically, enforcing this minimum sample size requirement filtered out regions with low data density and regions where species’ detections became rare.

L296 – what is meant by ‘within each base model’ here? Does it mean within each spatiotemporal block? Or the first time slice in each spatial block?

We have addressed the reviewer’s point by clarifying that “Within each spatiotemporal block, species’ occupancy and abundance were both assumed to be stationary.”

L299 – what kind of temporal trends? Within-year trends? Or trends across the years covered by the data?

Lines 358-361 provide an answer to this question, as follows: “The time of day was used to model variation in availability for detection; e.g., diurnal variation in behavior, such as participation in the “dawn chorus” Day of the year (1-366) was used to capture day-to-day changes in occupancy, and year was included to account for year-to-year differences.”

We have also clarified L346 by removing the word “trends” (because it is often interpreted to mean specifically inter-annual variation within conservation communities). This now reads “(ii) temporal predictors to account for temporal variation at various scales;”

L297 – some more detail is needed here on the BRT fitting within each spatiotemporal block (including the settings used eg bag fraction, learning rate, tree complexity etc). It would also be helpful to include a brief description of the types of relationships fitted by BRTs (linear / non-linear, interactions etc).

We have edited the paragraph to describe the base model, the occurrence and abundance boosted regression trees therein, and to provide more details about the boosted regression trees themselves. Specifically (Line 334): “Within each spatiotemporal block, species’ occupancy and abundance were both assumed to be stationary. If there were at least 50 checklists with at least 10 detections of the given species within the spatiotemporal block, we fit a two-step boosted regression tree model designed to deal with zero-inflation to predict the observed counts (abundance). If the minimum sample size requirements were not met for the spatiotemporal block, that spatiotemporal block was removed from the ensemble. The boosted regression trees for both steps of the zero-inflation model were fit with the gbm package with bag fraction = 0.80, learning rate or shrinkage = 0.05, and ntrees = 1000. The tree.depth parameter was set to 5 for the occurrence model and 10 for the abundance model, giving both models the ability to adapt to nonlinear and interacting predictor effects. We relied on the variance-reducing properties from averaging across the STEM ensemble to control for overfitting.”

We believe this addresses the reviewer’s comments but as with all comments, we remain open to additional suggestions for clarification.

L303 – more detail needed on what is meant by ‘spatial configuration’ here – how many configuration variables, and what type?

We have edited this section to provide more detail, as follows (Line 351): “All 19 cover classes in the MODIS data were summarized within 2.8×2.8 km (784 hectare) pixels centered at each eBird location. In each neighborhood, we computed the proportion of each class in the neighborhood (PLAND). To describe the spatial configuration of each class within each neighborhood we computed three statistics using using FRAGSTATS and SDMTools: LPI an index of the largest contiguous patch, PD an index of the patch density, and ED an index of the edge density.”

L316 – why 50-100 base models? What determined the number used?

The reviewer is correct that information regarding the ensemble design and how we used it was missing from the original text. We appreciate that point and apologize for the omission. To address this gap, we create the following brief description of the ensemble design (Line 321).

“The ensemble of spatiotemporal blocks was designed as a Monte Carlo sample of 100 randomly located spatiotemporal partitions of the study extent. This results a uniformly distributed set of spatiotemporal blocks, and up to 100 local models covering each location in the study extent. To account for spatial variation in the density of the bird observation data, smaller spatiotemporal blocks ($10^\circ \times 10^\circ \times 30$ continuous days) were used north of 12° latitude and larger blocks ($20^\circ \times 20^\circ \times 30$ continuous days) were used in the southern portion of the study extent.”

Second, we edited the end of the same section to explain how the number of spatiotemporal regions / base models was used and why we selected 50 as the minimum.

E.g. (Line 371), “Finally, to ensure good model performance, we required that the ensemble average computed for each relative abundance estimate have a sample size of at least 50 local models. In preliminary investigations based on expert review, we found that at least 50 local models were needed to control variation and limit extrapolation.”

L317 – what determined whether abundance or occupancy was used? How were abundance and occupancy outputs combined in the prioritization analysis?

We appreciate these questions and have edited the paragraph slightly by adding more detail to clarify how relative abundance was estimated.

Specifically (Line 365) “Relative abundance was estimated as the expected count of birds of a given species on a standardized, hypothetical search conducted by a typical eBird participant starting from the center of the pixel from 7:00 to 8:00 AM while traveling 1 km. Estimates of relative abundance were rendered at weekly temporal resolution and 8.4×8.4 km spatial resolution and computed as the product of the estimated occupancy and, the estimated abundance conditional on occupancy, from the two steps of the boosted regression tree model.”

Because the Method is described in detail in another published paper, we hope these details and that already included reference address these questions.

L316-318 – presumably the estimates of abundance for each species are each associated with a variance/uncertainty (propagated across the model ensemble), but these are essentially ‘ignored’ by taking the mean estimate forward in subsequent analysis. Was any model critique done to ensure that uncertainty levels were acceptable for each species & time slice?

The reviewer is correct - we do not use the uncertainty information about the relative abundance estimates in this analysis. It would be interesting to include this information in a later analysis, but doing so would dramatically increase computational effort (which is already substantial). We would very much like to consider such analyses in future, but suggest that it might be somewhat of a

distraction from our main points and thus have elected not to try and implement one of many possible approaches herein. However, we have considered this point and want to explain that rationale to be clear that we believe have indeed adequately accommodated uncertainty.

Specifically, in preliminary studies we observed that the number of local base-models available for the ensemble average, called ‘ensemble support’, provided a good way to control uncertainty and limit extrapolation for each 8.4 x 8.4km x 1 week pixel prediction. We also observed that uncertainty increased when regional data density and species detection rates declined, both of which affect ensemble support. We also evaluated model performance using expert review to set minimum regional sample size requirements (discussed above) to insure certainty in model outputs.

L326 & 329 – ensure not insure

Thank you; we have made the change.

L328 – presumably cluster analysis was done using the model predicted distribution maps, now raw abundance data – perhaps best to make this clear here.

Thank you: we have added ‘predicted’ to ‘weekly distribution maps’ to clarify.

L334 – In what circumstances were adjustments to the default 5 clusters made, and how often? Why not pick a specific cluster number for each species based on the known number of subspecies/subpopulations?

Thank you for this comment. We added text to the appropriate section in response to both reviewer 1 and 2 comments (Line 398):

“To do so, we first conducted cluster analyses of predicted weekly distribution maps for all 117 species to identify 5 clusters of equal abundance that encompassed the entire species range to ensure representation across it. We sampled populations as 5 clusters because population structure information was missing for most species, to facilitate a data driven spatial delineation of population centers, and for computational efficiency. However, we note that the appropriate number of clusters among species may vary by species phylogeography or abundance should sufficient information become available. Our clustering approach re-draws the clusters in each week of the year, so does not consider whether spatial structure is maintained across weeks – it simply imposes a plausible structure within each week.”

L337 - Please explain the rationale for using the two different cost metrics here (i.e. why the constant cost metric was used, given that all cells were assigned the same value). I guess the ‘constant cost’ metric was used for land-sharing, and the other one used for land-sparing, but this isn’t remotely clear. What was the threshold used for deciding whether a cell could be used in ‘land-sparing’ scenarios? Or was the human footprint used as a continuous constraint? The description of the differences between ‘sparing’ and ‘sharing’ need a much more detailed description.

Thank you, as judiciously as possible, we have added text to the ‘Land use constraints’ section of

Methods to describe more clearly the two cost metrics used in our paper, focusing especially on the human footprint metric (Line 419):

“We used the human footprint metric as the cost metric in low-impact scenarios. The human footprint layer represents a composite including the following human pressures: (1) the extent of built environments; (2) crop land; (3) pasture land; (4) human population density; (5) night-time lights; (6) railways; (7) roads; and (8) navigable waterways. The human footprint values range from 0 to 50, with 0 representing no human pressures and 50 representing the highest human pressure possible. We used the human footprint metric as a continuous constraint in the prioritization approach.”

Reviewers' Comments:

Reviewer #1:

Remarks to the Author:

The authors have done a very thorough job responding to reviewer comments. The descriptions of the approach to determine land areas necessary for meeting 30% population goals for the group of species when optimizing for each week of the year independently and summing the total area across all weeks ("weekly"), versus combining the data for all weeks over the entire annual cycle ("integrated annual") are clear. The conservation relevance has been enhanced by the fuller treatment of the conservation status and life history differences of the species analyzed (as well as indicating how these differences could relate in future analyses of other species groups). The real-world conservation relevance and contribution of the paper has been enhanced by the treatment of trade-offs for conservation practitioners in optimizing conservation goals, cost and feasibility. I feel that the points raised in the previous round of review have been satisfactorily addressed.

Reviewer #2:

Remarks to the Author:

The authors have done a good job of revising their manuscript in response to both reviews. I have a few remaining concerns, which relate almost entirely to the wording of the manuscript and terms used for key concepts / scenarios.

Firstly, I think the new term used for the annual prioritization approach, 'integrated annual', is still rather confusing. The new revision does a much better job of explaining the difference between weekly and annual analyses, but I think the wording in the abstract (and thereafter) will be confusing to readers – i.e. 'annually integrated across all weeks...(L31)'.

My understanding (which may still be incorrect) is that the integrated annual prioritization is essentially the same thing as summing the 52 weekly abundance maps together to give a 'total annual abundance' map, and then running a prioritisation on that map to select cells capturing 30% of the summed annual abundance. This would presumably mean that a cell that is occupied at low abundance for most of the year could be more likely to be selected than a cell that supports a very high abundance for only one week of the year.

I think some wording is needed in the main manuscript (early on) that makes this distinction clear – calling it 'integrated annual' is still too vague. Ideally, the wording should not only explain what the integrated annual approach does (prioritization of summed weekly abundances), but also explains the rationale for why this approach is being considered at all. The idea seems to be to prioritize cells that are occupied more consistently across the annual cycle, as opposed to the weekly approach which prioritizes cells that hold high abundance at any point in the annual cycle. It would be useful to provide some explanation (and ideally subsequent discussion) of the biological rationale for these two approaches, in terms of what we know about migratory population dynamics. The 'integrated annual' approach will presumably give a higher guarantee that sites important for breeding and wintering are protected (ie periods of static occupancy), whilst the weekly approach gives a higher guarantee that stop-over sites are protected, but potentially at the expense of some breeding/winter habitat. Some further discussion of how the two approaches could differ in terms of their outcomes for population dynamics is warranted.

I also struggle with the new term for the population clustering analysis – 'spatial population abundance variation'. I acknowledge that this term is being used in response to a point I made in my

previous review, and I agree that it's an improvement on the previous wording. However, again I think the precise term being used might be confusing to readers, because 'spatial population abundance variation' could be taken to imply continuous variation in abundance, rather than patchiness in distributions (which is what this analysis is ultimately about). Perhaps a good term would be something like 'spatial clustering of species abundance, or 'spatial patchiness of species abundance'?

The sentence at L72 could then be reworded thus:

"...by sampling species a) over their entire range, without accounting for spatial clustering of species abundance, or b) by sampling within 5 regional population clusters, identified weekly to accommodate spatial structure in population abundance".

This should then be followed by a brief sentence explaining why it is important to take this kind of clustering into account – i.e. the fact that species distributions are usually patchy, and conservation should ideally ensure that multiple 'patches' are adequately protected to avoid putting all eggs in one basket...

To further clarify what the clustering does, I think it would also be good if the supplement could include a figure showing an example of the outcome of clustering for a single species in a single week – i.e. an abundance map with the clusters delineated over the top. This would help to visually illustrate the sort of results the method generates.

Specific comments, primarily on clarity of wording:

L31 – Does the non-structured version genuinely 'ignore' spatial variation in abundance? Abundance is still considered in the analysis, just not the 5-level spatial clustering. Wording might need tweaking for clarity.

L33 – I appreciate that it's tricky with severe Abstract word constraints, but the Abstract currently lacks any statement of the actual results – if possible, it would be good to change the final sentence to a summary of the actual direction of trade-offs (rather than simply a statement that there were trade-offs).

L78 – it's important to state here that they are equal-abundance clusters (i.e. they don't lead to a spatially isolated 'cluster' that contains only 5% of the population being given equal weighting to another that contains 75%).

L86 – I appreciate that the new terms coined for the land-use scenarios were in response to my own comments in the previous review, but I don't think 'low-impact' is a very good term for the scenario that limits conservation to intact habitat. Low impact on what? Why not simply call it the 'intact habitat' scenario? That has a straightforward interpretation in comparison to the 'shared-use' scenario.

L99 – I still don't really follow this sentence:

"Area reductions under annually-integrated planning generally resulted from cases such as the inclusion of sites used by more than one species across two or more weeks of the annual cycle."

I think my confusion about this also relates to the point I make above about clearly explaining the biological implications of the annual vs weekly approaches. Perhaps the wording needs to emphasise

that the annual approach will select sites that are used for longer periods of the annual cycle, and that there may be greater overlap of those areas between species than occurs in the short-term stopover sites included in the weekly approach, hence the larger area needed under the latter. I appreciate it's difficult to explain these complex results in few words, but I think a bit more tweaking of the current wording is needed to make things clearer to the reader.

L104 – If possible, it would be good to briefly propose an explanation for why the clustering had less of an impact on the intact-habitat scenario compared to shared-use – it's not immediately obvious why this would be the case.

L118 – Perhaps worth noting that 'efficiency of conservation' here is specifically about migrants – the efficiency of selecting shared-use areas may be far lower for the local biodiversity in those areas (indeed this is usually the case in the tropics).

L128 – I suggest the following sentence needs to be reworded, as it almost reads like you're advocating using less precise methods: "Our findings suggest a need to re-evaluate conservation planning processes based on less precise methods."

L158-174 – Whilst this paragraph is interesting, it could be drastically shortened without too much loss of content in order to accommodate space within the word limit to better explain the rationale and methods. Same goes for the Conclusion paragraph.

Figure 1 - There's still some confusing text on the right-hand maps which needs removing.

Please find our responses to reviewer comments in bold below. Line numbers included refer to lines in text without tracked changes.

REVIEWERS' COMMENTS:

Reviewer #1 (Remarks to the Author):

The authors have done a very thorough job responding to reviewer comments. The descriptions of the approach to determine land areas necessary for meeting 30% population goals for the group of species when optimizing for each week of the year independently and summing the total area across all weeks (“weekly”), versus combining the data for all weeks over the entire annual cycle (“integrated annual”) are clear. The conservation relevance has been enhanced by the fuller treatment of the conservation status and life history differences of the species analyzed (as well as indicating how these differences could relate in future analyses of other species groups). The real-world conservation relevance and contribution of the paper has been enhanced by the treatment of trade-offs for conservation practitioners in optimizing conservation goals, cost and feasibility. I feel that the points raised in the previous round of review have been satisfactorily addressed.

Thank you very much for this encouraging comment. We very much appreciate your help.

Reviewer #2 (Remarks to the Author):

The authors have done a good job of revising their manuscript in response to both reviews. I have a few remaining concerns, which relate almost entirely to the wording of the manuscript and terms used for key concepts / scenarios.

Firstly, I think the new term used for the annual prioritization approach, ‘integrated annual’, is still rather confusing. The new revision does a much better job of explaining the difference between weekly and annual analyses, but I think the wording in the abstract (and thereafter) will be confusing to readers – i.e. ‘annually integrated across all weeks...(L31)’.

Thank you very much for this comment. We have decided to use the terms ‘weekly’ and ‘yearly’ as the terms representing the two different approaches. We hope this helps reduce confusion.

We have changed the wording in question from the abstract to: ‘for each week of the year independently vs. combined’

We further went ahead and changed ‘integrated annual’ to ‘yearly’ throughout the text.

My understanding (which may still be incorrect) is that the integrated annual prioritization is essentially the same thing as summing the 52 weekly abundance maps together to give a ‘total annual abundance’ map, and then running a prioritisation on that map to select cells capturing 30% of the summed annual abundance. This would presumably mean that a cell that is occupied at low abundance for most of the year could be more likely to be selected than a cell that supports a very high abundance for only one week of the year.

Thank you very much for this comment. We have added text to the methods section to further clarify the distinction between weekly and yearly approaches (line 355):

“For the yearly approach, each week of the year is its own feature in the optimization. This means, that for a weekly scenario we would have 117 features (one for each species) and in the yearly 117 * 52 features (one for each species in each week of the year).”

The meaning of the differences is illustrated further in Supplementary Figure 6, as well as in the Methods section (line 362):

“The example shows that in general the weekly approach captures areas important for each week of the year, independent of other weeks of the year. The ‘yearly’ approach on the other hand optimizes across the year, thereby minimizing area requirements, but as a trade-off it does not necessarily capture the most important areas in each week of the year.”

I think some wording is needed in the main manuscript (early on) that makes this distinction clear – calling it ‘integrated annual’ is still too vague. Ideally, the wording should not only explain what the integrated annual approach does (prioritization of summed weekly abundances), but also explains the rationale for why this approach is being considered at all. The idea seems to be to prioritize cells that are occupied more consistently across the annual cycle, as opposed to the weekly approach which prioritizes cells that hold high abundance at any point in the annual cycle. It would be useful to provide some explanation (and ideally subsequent discussion) of the biological rationale for these two approaches, in terms of what we know about migratory population dynamics. The ‘integrated annual’ approach will presumably give a higher guarantee that sites important for breeding and wintering are protected (ie periods of static occupancy), whilst the weekly approach gives a higher guarantee that stop-over sites are protected, but potentially at the expense of some breeding/winter habitat. Some further discussion of how the two approaches could differ in terms of their outcomes for population dynamics is warranted.

Thank you very much for this comment. We have reworded the first mention of the terms in the Introduction now and added an explanatory sentence for each approach to more clearly define their meanings (Line 66):

“We next recorded and compared the geographic area requirements and land cover types selected when optimizing for each week of the year independently and summing the total area over all weeks (hereafter, ‘weekly’), versus optimizing over the entire year at once (hereafter, ‘yearly’). Weekly optimizations for area efficiencies were developed to identify species-specific priorities for species at fine enough scales to capture short-term stopover sites. Our yearly approach optimized efficiency over the full annual cycle of each species, and emphasized temporal consistency in abundance hotspots more likely to reflect breeding and non-breeding regions.”

We discuss the outcomes of the approaches in the Discussion section at the end of the paper and include this text here for reference, as we think it sufficiently addresses the reviewer’s comment (line 205):

“First, scenarios based on the distributions of abundance of all 117 species that were integrated over the entire year required less land area to meet conservation targets than scenarios based on optimizations that used the weekly distributions of those species throughout the year. This suggests difficult trade-offs between finer-scale spatiotemporal representation of habitats throughout the year, versus solutions that achieve the best compromise on population targets across the entire year. The latter appears more area-efficient, but may miss key stopover points along migratory routes.”

I also struggle with the new term for the population clustering analysis – ‘spatial population abundance

variation'. I acknowledge that this term is being used in response to a point I made in my previous review, and I agree that it's an improvement on the previous wording. However, again I think the precise term being used might be confusing to readers, because 'spatial population abundance variation' could be taken to imply continuous variation in abundance, rather than patchiness in distributions (which is what this analysis is ultimately about). Perhaps a good term would be something like 'spatial clustering of species abundance, or 'spatial patchiness of species abundance'?

Thank you very much for this comment. We have changed the wording to 'spatial clustering of species abundance' throughout the text now.

The sentence at L72 could then be reworded thus:

"...by sampling species a) over their entire range, without accounting for spatial clustering of species abundance, or b) by sampling within 5 regional population clusters, identified weekly to accommodate spatial structure in population abundance".

This should then be followed by a brief sentence explaining why it is important to take this kind of clustering into account – i.e. the fact that species distributions are usually patchy, and conservation should ideally ensure that multiple 'patches' are adequately protected to avoid putting all eggs in one basket...

Thank you very much for this suggestion. We have now added a text to address this point (Line 80): "We wanted to account for the spatial clustering of species abundance because broadly distributed species often exhibit strong regional-scale variation in abundance across their range. Regional-scale variation in species' abundance may reflect a number of important processes affecting the ecology and conservation of species, from variation in resource availability and land-use patterns to population-structure related to movement and migratory connectivity. By accounting for the spatial clustering of species abundance, the prioritization is stratified over multiple regions to ensure adequate protection over the entire species range."

To further clarify what the clustering does, I think it would also be good if the supplement could include a figure showing an example of the outcome of clustering for a single species in a single week – i.e. an abundance map with the clusters delineated over the top. This would help to visually illustrate the sort of results the method generates.

Thank you very much for this suggestion. We have now produced the figure below and added it to Supplementary Information

Supplementary Information Figure 6. Example outcome of clustering approach. For this example, we used Canada Warbler as the species and week of June 6. Figure a) shows the relative abundance of that species in that week of the year, and b) shows the output from the clustering analysis. Each color in b) corresponds to one cluster that was produced using the CLARA algorithm, which is an extension of the k-medoids technique for clustering of large datasets. Each of the clusters in b) was treated as a separate feature (using relative abundance values from a) in scenarios where clustered datasets were used.

Specific comments, primarily on clarity of wording:

L31 – Does the non-structured version genuinely ‘ignore’ spatial variation in abundance? Abundance is still considered in the analysis, just not the 5-level spatial clustering. Wording might need tweaking for clarity.

We have changed the wording to the reviewers suggestion ‘spatial clustering of species abundance’, which we believe addressed the wording issue.

L33 – I appreciate that it’s tricky with severe Abstract word constraints, but the Abstract currently lacks

any statement of the actual results – if possible, it would be good to change the final sentence to a summary of the actual direction of trade-offs (rather than simply a statement that there were trade-offs).

Thank you very much for this comment. We have reworded the abstract to reflect this suggestion. The abstract now reads:

“Limited knowledge of the distribution, abundance, and habitat associations of migratory species hinders effective conservation actions. We use Neotropical migratory birds as a model group to compare approaches to prioritize land conservation needed to support $\geq 30\%$ of the global abundances of 117 species. Specifically, we compare scenarios from spatial optimization models to achieve conservation targets by: 1) area requirements for conserving $>30\%$ abundance of each species for each week of the year independently vs. combined; 2) including vs. ignoring spatial clustering of species abundance; and 3) incorporating vs. avoiding human-dominated landscapes. Solutions integrating information across the year require 50% less area than those integrating weekly abundances, with additional reductions when shared-use landscapes are included. Although incorporating spatial population structure requires more area, geographical representation among priority sites improves substantially. These findings illustrate that globally-sourced citizen science data can elucidate key trade-offs among opportunity costs and spatiotemporal representation of conservation efforts.”

L78 – it’s important to state here that they are equal-abundance clusters (i.e. they don’t lead to a spatially isolated ‘cluster’ that contains only 5% of the population being given equal weighting to another that contains 75%).

Thank you for this comment. We modified text to read: ‘.. , we used cluster analysis to delineate abundances into 5 spatial clusters of equal abundance..’

L86 – I appreciate that the new terms coined for the land-use scenarios were in response to my own comments in the previous review, but I don’t think ‘low-impact’ is a very good term for the scenario that limits conservation to intact habitat. Low impact on what? Why not simply call it the ‘intact habitat’ scenario? That has a straightforward interpretation in comparison to the ‘shared-use’ scenario.

Thank you for this comment. We changed ‘low-impact’ to ‘intact habitat’ throughout the text and supplementary material.

L99 – I still don’t really follow this sentence:

“Area reductions under annually-integrated planning generally resulted from cases such as the inclusion of sites used by more than one species across two or more weeks of the annual cycle.”

I think my confusion about this also relates the point I make above about clearly explaining the biological implications of the annual vs weekly approaches. Perhaps the wording needs to emphasise that the annual approach will select sites that are used for longer periods of the annual cycle, and that there may be greater overlap of those areas between species than occurs in the short-term stopover

sites included in the weekly approach, hence the larger area needed under the latter. I appreciate it's difficult to explain these complex results in few words, but I think a bit more tweaking of the current wording is needed to make things clearer to the reader.

Thank you very much for this comment. We have incorporated the suggestion by the reviewer and added the following sentence (Line 114):

“A likely explanation for this difference is that the yearly approach will select sites that are used for longer periods of the annual cycle, and that there may be greater overlap of those areas between species than occurs in the short-term stopover sites included in the weekly approach, hence the larger area needed under the latter.”

L104 – If possible, it would be good to briefly propose an explanation for why the clustering had less of an impact on the intact-habitat scenario compared to shared-use – it's not immediately obvious why this would be the case.

Thank you for this comment. We have added the following sentence to propose an explanation (Line 123):

“This reduction occurred in part because the homogenous cost structure used in our intact habitat scenario was less influential on site selection than was the heterogenous cost structure used in the shared-use scenario.”

L118 – Perhaps worth noting that 'efficiency of conservation' here is specifically about migrants – the efficiency of selecting shared-use areas may be far lower for the local biodiversity in those areas (indeed this is usually the case in the tropics).

We have moved migratory species to earlier in the sentence to make clear efficiency of conservation is related to migrants:

“efficiency of conservation area designs for migrants if their demographic performance is similar in 'working' and 'intact' landscapes.”

L128 – I suggest the following sentence needs to be reworded, as it almost reads like you're advocating using less precise methods: “Our findings suggest a need to re-evaluate conservation planning processes based on less precise methods.”

Thank you for this comment. The sentence now reads:

“Our findings suggest a need to re-evaluate conservation planning processes that are based on less precise methods.”

L158-174 – Whilst this paragraph is interesting, it could be drastically shortened without too much loss of content in order to accommodate space within the word limit to better explain the rationale and methods. Same goes for the Conclusion paragraph.

We very much appreciate this comment, but given that we are well below the word limit and think that both the paragraph in question and the conclusion paragraph (renamed to Discussion to comply with journal policies) hold valuable information, we opted to leave both sections intact.

Figure 1 - There's still some confusing text on the right-hand maps which needs removing.

We have now removed the text from the right-hand maps.